# Sleep Quality and Urinary Incontinence in Prostate Cancer Patients: A Data Analytics Approach with the ASCAPE Dataset

**DOI:** 10.3390/healthcare12181817

**Published:** 2024-09-11

**Authors:** Ioannis Manolitsis, Georgios Feretzakis, Lazaros Tzelves, Athanasios Anastasiou, Yiannis Koumpouros, Vassilios S. Verykios, Stamatios Katsimperis, Themistoklis Bellos, Lazaros Lazarou, Ioannis Varkarakis

**Affiliations:** 1Second Department of Urology, Sismanoglio General Hospital, National and Kapodistrian University of Athens, 15126 Athens, Greece; lazarostzelves@gmail.com (L.T.); stamk1992@gmail.com (S.K.); bellosthemistoklis@gmail.com (T.B.); lazarou_laz@hotmail.com (L.L.); medvark3@yahoo.com (I.V.); 2School of Science and Technology, Hellenic Open University, 26335 Patras, Greece; georgios.feretzakis@ac.eap.gr (G.F.); verykios@eap.gr (V.S.V.); 3Biomedical Engineering Laboratory, National Technical University of Athens, 15780 Athens, Greece; aanastasiou@biomed.ntua.gr; 4Digital Innovation in Public Health Research Laboratory, Department of Public and Community Health, University of West Attica, 11521 Athens, Greece; ykoump@uniwa.gr

**Keywords:** artificial intelligence, quality of life, prostate cancer, sleep quality, urinary incontinence

## Abstract

Background: The ASCAPE project aims to improve the health-related quality of life of cancer patients using artificial intelligence (AI)-driven solutions. The current study employs a comprehensive dataset to evaluate sleep and urinary incontinence, thus enabling the development of personalized interventions. Methods: This study focuses on prostate cancer patients eligible for curative treatment with surgery. Forty-two participants were enrolled following their diagnosis and were followed up at baseline and 3, 6, 9, and 12 months after surgical treatment. The data collection process involved a combination of standardized questionnaires and wearable devices, providing a holistic view of patients’ QoL and health outcomes. The dataset is systematically organized and stored in a centralized database, with advanced statistical and AI techniques being employed to reveal correlations, patterns, and predictive markers that can ultimately lead to implementing personalized intervention strategies, ultimately enhancing patient QoL outcomes. Results: The correlation analysis between sleep quality and urinary symptoms post-surgery revealed a moderate positive correlation between baseline insomnia and baseline urinary symptoms (r = 0.407, *p* = 0.011), a positive correlation between baseline insomnia and urinary symptoms at 3 months (r = 0.321, *p* = 0.049), and significant correlations between insomnia at 12 months and urinary symptoms at 3 months (r = 0.396, *p* = 0.014) and at 6 months (r = 0.384, *p* = 0.017). Furthermore, modeling the relationship between baseline insomnia and baseline urinary symptoms showed that baseline insomnia is significantly associated with baseline urinary symptoms (coef = 0.222, *p* = 0.036). Conclusions: The investigation of sleep quality and urinary incontinence via data analysis through the ASCAPE project suggests that better sleep quality could improve urinary disorders.

## 1. Introduction

The advancements that have been made in cancer detection and treatment procedures have led to an increase in the number of patients who have outlived cancer [1]. It has been established that prostate cancer is the second most prevalent neoplasm among men, accounting for around 336,000 new cases each year in Europe alone [2]. Cancer patients frequently experience adverse effects from the disease or its treatment, which can substantially diminish their health-related quality of life (QoL) [3]. Because of the shortcomings in healthcare and financial resources, in addition to a restricted personalized-based approach in the rehabilitation plan, the current strategy for supporting cancer patients and survivors does not satisfy their needs. This is mainly due to the fact that the rehabilitation plan is simply not sufficiently tailored to individual patients [4,5,6].

As a result of developments in Big Data analytics and artificial intelligence (AI), numerous aspects of healthcare have been made more accessible for improvement. AI is increasingly being utilized in oncology, both in the diagnostic and screening processes [7]. The ASCAPE (Artificial Intelligence Supporting Cancer Patients throughout Europe) project is a large-scale, multicentric, prospective initiative with the goal of improving the quality of life (QoL) of patients suffering from breast and prostate cancer through the use of AI-driven therapies [8]. A comprehensive dataset comprising information gathered from patients with prostate cancer who have undergone radical prostatectomy (RP) is the primary focus of this investigation. For the purpose of studying various elements of quality of life, such as the quality of sleep and urinary incontinence (UI), this dataset is essential.

Using this extensive dataset, the ASCAPE project intends to discover important insights into the relationship between the quality of sleep that prostate cancer patients experience and the quality of voiding that they experience during their convalescence period. Consequently, this can provide information for artificial intelligence models that are intended to forecast and improve patient outcomes, which can ultimately result in individualized interventions that improve the quality of life of cancer patients.

## 2. Methods

### 2.1. General Design

The ASCAPE study was designed as a large-scale, non-invasive, multicentric, and prospective research initiative that spans multiple centers across Europe. The purpose of ASCAPE (Artificial intelligence Supporting CAncer Patients across Europe) is to use the latest developments in Big Data and AI (Artificial Intelligence) to support the quality of life and health status of cancer patients. This project involves 15 partners from 7 countries, including academic medical centers, SMEs (small and medium-sized enterprises), research centers, and universities.

This comprehensive study leverages advanced artificial intelligence to enhance the quality of life (QoL) of prostate cancer patients. By integrating data from numerous participants across diverse geographical locations, this study aims to generate a robust dataset that can be used to train AI models for personalized healthcare interventions [8]. This study’s non-invasive nature ensures minimal disruption to the participants’ daily lives while enabling continuous and detailed monitoring of their health status over an extended period. The ASCAPE project has been registered on the database of clinicaltrials.gov under the identifier NCT04879563.

### 2.2. Participants

The participant cohort for the ASCAPE study consists of patients who have been diagnosed with prostate cancer. These patients undergo open RP as part of their treatment regimen. The inclusion criteria are stringent to ensure the relevance and quality of the data collected. We included prostate cancer patients without clinical evidence of metastatic disease, eligible for curative treatment with surgery. Patients were selected based on specific clinical and demographic criteria, ensuring a diverse yet representative sample of the broader prostate cancer patient population [9]. Signed informed consent before study inclusion was mandatory for all patients.

Forty-two participants were enrolled in this study following their diagnosis. The follow-up period was structured to include periodic assessments at baseline and at 3, 6, 9, and 12 months post treatment. These follow-ups were critical for tracking the progression of patients’ QoL and health outcomes over time. Each assessment phase involved comprehensive data collection to monitor various health parameters, psychological well-being, and overall QoL.

### 2.3. Data Collection

The data collection process in the ASCAPE study is multi-faceted, involving a combination of standardized questionnaires and wearable devices. This dual approach ensures the holistic capture of both subjective and objective health metrics.

The dataset used in this study contained detailed information collected through standardized questionnaires and wearable devices over a 12-month period post-surgery. The questionnaires used included the EORTC QLQ PR25 for urinary symptoms and general health [10], the HADS for anxiety and depression [11], and the IIEF for erectile function [12]. These questionnaires were administered at baseline and 3, 6, 9, and 12 months post-surgery, providing a longitudinal view of the patients’ QoL.

In addition to questionnaire data, the dataset included extensive demographic and clinical features such as age, presenting symptoms, comorbidities, drug regimen, family history of prostate cancer, smoking history, marital status, weight, height, working status, living location, PSA levels pre-operation, Gleason scores, surgical margins, blood loss during surgery, length of hospital stay, duration of surgery, biochemical recurrence, and risk group classification.

The dataset’s richness was further enhanced by the inclusion of sleep data from Fitbit watches, which tracked patients’ sleep patterns, heart rate variability, and movement. This allowed for a more granular analysis of sleep quality and its relationship with UI and other health outcomes [13].

#### 2.3.1. Standardized Questionnaires

EORTC QLQ PR25 (European Organization for Research and Treatment of Cancer Quality of Life Questionnaire—Prostate Cancer Module): This questionnaire assesses specific QoL issues related to prostate cancer, including urinary symptoms, bowel symptoms, treatment-related side effects, and overall health status.HADS (Hospital Anxiety and Depression Scale): This scale measures the levels of anxiety and depression among patients, providing insights into their mental health status.IIEF (International Index of Erectile Function): This questionnaire evaluates erectile function, which is a critical aspect of QoL for prostate cancer patients.

#### 2.3.2. Wearable Devices

Fitbit Watches: These devices were provided to participants to continuously monitor their daily activity levels, sleep patterns, heart rate variability, and other vital health metrics. The wearable data complemented the subjective questionnaire responses, offering a comprehensive view of the patients’ health and QoL.

The combination of these data sources allowed for a detailed and nuanced understanding of the patients’ health trajectories. The wearable devices provided real-time, objective data that could capture fluctuations and trends that may not be evident through periodic questionnaires alone.

### 2.4. Data Management and Analysis

The collected data were systematically organized and stored in a centralized database, ensuring data integrity and accessibility for analysis. Advanced statistical and AI techniques were employed to analyze the data, uncovering correlations, patterns, and predictive markers that can inform personalized intervention strategies. The AI models were trained on this rich dataset to provide tailored recommendations for improving QoL, addressing specific health issues identified through the analysis. This study also emphasized data privacy and security, ensuring that all patient data were anonymized and handled in compliance with relevant data protection regulations [14]. The insights derived from this study were used to refine and enhance the AI models continuously, improving their accuracy and effectiveness in recommending interventions for future patients.

#### 2.4.1. Data Analysis

The analysis in this study primarily focused on understanding the correlation between sleep quality and UI in prostate cancer patients. Additionally, it examined the demographic and clinical features of these patients to provide a comprehensive view of the factors influencing their QoL post treatment. The dataset used for this analysis was explored using Python and relevant statistical methods, ensuring a rigorous and systematic approach to uncovering key insights [15,16].

#### 2.4.2. Data Preparation

The dataset was initially loaded from an Excel file and reviewed to understand its structure and contents. Key columns related to sleep quality and UI were identified and extracted. The selected columns for sleep quality included measurements taken at baseline and 3 months, 6 months, 9 months, and 12 months post treatment. Similarly, columns representing urinary incontinence symptoms at corresponding time points were selected. To ensure the integrity of the analysis, any rows containing missing values within these selected columns were removed. This step was crucial to avoid biases or inaccuracies in the subsequent correlation analysis.

#### 2.4.3. Correlation Analysis

A correlation matrix was computed to examine the relationships between sleep quality and UI scores. This matrix allowed the identification of specific time points where these two health aspects might be significantly related. The correlation coefficients indicated the strength and direction of these relationships, providing a quantitative measure of how changes in sleep quality were associated with changes in UI symptoms. In order to complement the correlation analysis, *p*-values were calculated for each correlation coefficient. This statistical measure helped to determine the significance of the observed correlations, ensuring that the results were not due to random chance. By applying Pearson correlation tests, the analysis provided a robust understanding of the interactions between sleep quality and UI over time.

What is more, in order to investigate the relationship between anxiety and urinary incontinence (UI), we calculated Pearson’s correlation coefficients between anxiety scores, as measured by the HADS (Hospital Anxiety and Depression Scale) at different time points (baseline, 3, 6, 9, and 12 months), and urinary symptoms, measured by the QLQ-PR25 questionnaire at corresponding time points. Data from both anxiety and UI scores were extracted, and rows with missing data were excluded from the analysis. Statistical significance was determined with a *p*-value threshold of 0.05.

#### 2.4.4. Subgroup Analysis

We performed a subgroup analysis to explore the relationships between urinary symptoms and erectile function across different age groups. Patients were divided into three age categories: <60 years, 60–70 years, and >70 years. For each subgroup, we calculated the Pearson correlation coefficient between baseline urinary symptoms, measured using the QLQ-PR25 questionnaire, and baseline erectile function, measured using the IIEF questionnaire. This analysis allowed us to examine potential variations in the relationship between these outcomes across age groups. Furthermore, we applied a random forest regression model to predict baseline urinary symptoms using age, comorbidities, and preoperative PSA levels as predictors. The data were split into training and test sets, and the model was evaluated using the mean squared error (MSE) and R-squared (R^2^). Feature importance was also calculated to assess the relative contribution of each predictor to the model’s performance.

#### 2.4.5. Demographic and Clinical Features

Beyond the primary focus on sleep quality and UI, the analysis extended to the demographic and clinical features of the patients. Key demographic data such as age, weight, height, and smoking history were examined alongside clinical variables like PSA levels, Gleason scores, and the duration of surgery. These variables were summarized to offer a detailed profile of the patient population, enabling a more nuanced interpretation of the primary correlation findings. Descriptive statistics were generated for these demographic and clinical features, providing insights into the average values, variability, and distributions within the patient cohort. This comprehensive summary helped to contextualize the correlation results, suggesting potential confounding factors or additional variables that might influence the observed relationships.

#### 2.4.6. Statistical Tests

Several statistical tests were employed to validate and extend the findings from the correlation analysis. A paired *t*-test was used to compare baseline and 3-month sleep quality scores, assessing whether significant changes occurred post treatment. An ANOVA test was conducted to evaluate differences in sleep quality scores across all time points, exploring whether sleep quality varied significantly throughout the follow-up period [17,18]. Additionally, a chi-square test examined the independence between smoking history and comorbidities, investigating whether these demographic factors were associated. Finally, a linear regression analysis was performed to model the relationship between baseline sleep quality and baseline urinary symptoms, providing an in-depth look at how initial sleep quality might predict urinary health outcomes [19].

For the purposes of further clarifying and visually representing our methodology, we present a block diagram. This diagram outlines the key steps of our workflow (Figure 1), from data collection and preprocessing to statistical analysis, machine learning modeling, and interpretation of results. Thus, we provide a clear and concise visual overview of our approach, highlighting each stage of the process and our contributions in a structured manner.

## 3. Results

### 3.1. Demographic and Clinical Features

Table 1 summarizes the demographic and clinical characteristics of the patient cohort, providing context for the correlation analysis. Furthermore, Table 2 illustrates the time-dependent variables of this study (urinary symptoms and erectile function), along with key information about our patients. We included 42 patients, and the average age of participants was 65.1 years, with the majority presenting with lower urinary tract symptoms (LUTSs). Key clinical variables included PSA levels, Gleason score, and comorbidities, which are crucial for understanding the patient profiles.

The analysis of the ASCAPE dataset involved examining the correlation between sleep quality and UI, alongside demographic and clinical features of prostate cancer patients who underwent RP. The dataset, comprising various questionnaires and clinical measurements, provided a comprehensive view of patient health over a 12-month period post-surgery.

### 3.2. Correlation Analysis

The correlation matrix (Figure 2) reveals the relationships between sleep quality and urinary symptoms across different time points. The key findings include the following:A moderate positive correlation between baseline insomnia and baseline urinary symptoms (r = 0.407, *p* = 0.011).A positive correlation between baseline insomnia and urinary symptoms at 3 months (r = 0.321, *p* = 0.049).Significant correlations between insomnia at 12 months and urinary symptoms at 3 months (r = 0.396, *p* = 0.014) and at 6 months (r = 0.384, *p* = 0.017).

These correlations suggest that poorer sleep quality is associated with more severe urinary symptoms, indicating a potential area for targeted interventions.

The Pearson correlation analysis between anxiety and urinary incontinence (UI) revealed weak to moderate correlations at various time points. While most correlations were not statistically significant, we observed a statistically significant moderate correlation between baseline anxiety and urinary symptoms at 12 months (r = 0.46, *p* = 0.0037). Additionally, there was a significant positive correlation between anxiety at 3 months and urinary symptoms at 6 months (r = 0.34, *p* = 0.0373). Other correlations, although present, did not reach statistical significance. Table 3 highlights the statistically significant correlations between anxiety scores and urinary incontinence, with the most notable associations observed between baseline anxiety and urinary symptoms at 12 months, as well as anxiety at 3 months and urinary symptoms at 6 months.

A large-scale cross-sectional study from the Norwegian HUNT study explored the association between anxiety, depression, and urinary incontinence (UI) in the general female population. Felde et al. (2020) demonstrated a significant relationship between these variables, with moderate to severe anxiety or depression increasing the prevalence of UI by 37% to 57%. Mixed UI was most strongly associated with high anxiety and depression scores. Notably, the use of antidepressants was associated with increased odds of UI, while the use of anxiolytics was associated with decreased odds after adjusting for anxiety [20]. While these findings are from a different population than our study of prostate cancer patients, they highlight the complex interplay between psychological factors, medication use, and urinary symptoms. This relationship warrants further investigation in the context of prostate cancer patients, particularly given the potential psychological impact of cancer diagnosis and treatment.

### 3.3. Subgroup Analysis

The subgroup analysis revealed age-related variations in the correlation between baseline urinary symptoms and erectile function. In patients younger than 60 years, there was a moderate negative correlation (r = −0.42), indicating that more severe urinary symptoms were associated with worse erectile function. Similarly, in the 60–70 age group, a weaker negative correlation was observed (r = −0.25). Interestingly, in patients older than 70 years, we found a positive correlation (r = 0.41), suggesting that in this subgroup, worse urinary symptoms were unexpectedly associated with better erectile function. These findings highlight potential age-related differences in the relationship between urinary and sexual health outcomes following prostate cancer treatment.

The random forest regression model achieved a mean squared error (MSE) of 211.24 and an R-squared (R^2^) value of 0.22, indicating that the model explained 22% of the variability in baseline urinary symptoms. The preoperative PSA level emerged as the most important predictor, with a feature importance score of 0.54, followed by age (0.34) and comorbidities (0.13). These results suggest that preoperative PSA levels and age play significant roles in predicting post-surgical urinary symptoms.

### 3.4. Statistical Tests

Several statistical tests were performed to validate the findings from the correlation analysis:Paired *t*-Test: Comparing baseline and 3-month insomnia scores yielded a *t*-statistic of 1.653 and a *p*-value of 0.107, indicating no significant change in sleep quality over this period.ANOVA: Analyzing insomnia scores across all time points (baseline, 3, 6, 9, and 12 months) resulted in an F-statistic of 0.987 and a *p*-value of 0.416, suggesting no significant differences in insomnia scores over time.Chi-Square Test: Examining the independence between smoking history and comorbidities produced a chi^2^ value of 16.648 and a *p*-value of 0.340, indicating no significant association between these variables.Linear Regression: Modeling the relationship between baseline insomnia and baseline urinary symptoms showed that baseline insomnia is significantly associated with baseline urinary symptoms (coef = 0.222, *p* = 0.036).

### 3.5. Visualization of Results

The box plot illustrated in Figure 3 shows a downswing in the levels of anxiety for the patients over a 12-month period of follow up after receiving surgical treatment for localized prostate cancer. What is more, Figure 4 presents the depression levels of our patients for the same time period, showing a steady decline from baseline (pre-surgery) to the 12-month landmark.

The diagram shown in Figure 5 indicates a concentration around the mean age of 65 years. This in accordance with global statistics for prostate cancer which show that the average age of men diagnosed with prostate cancer is around 66 years [1,2].

The results of the ASCAPE study highlight important correlations between sleep quality and UI in prostate cancer patients post-surgery. The analysis underscores the potential impact of poor sleep on urinary symptoms, suggesting avenues for targeted interventions. The integration of demographic and clinical data enriches our understanding of these relationships, offering a comprehensive view of patient health trajectories. Statistical tests and visualizations provide robust support for the findings, guiding future research and clinical practices in improving the quality of life for cancer patients.

## 4. Discussion

This study provides a thorough examination of the relationship between the quality of sleep and UI in prostate cancer patients who have undergone RP. This research provides valuable insights into the QoL after surgery for these patients by using a comprehensive dataset that includes several health and demographic parameters collected at multiple time periods.

The subgroup analysis revealed notable age-related variations in the correlation between baseline urinary symptoms and erectile function. In patients younger than 60 years, a moderate negative correlation (r = −0.42) was observed, suggesting that higher urinary symptoms were associated with worse erectile function. A similar but weaker negative correlation (r = −0.25) was found in the 60–70 age group. Interestingly, patients older than 70 exhibited a positive correlation (r = 0.41), indicating that worse urinary symptoms were unexpectedly associated with better erectile function in this subgroup. These findings suggest that age-related differences may play a significant role in the relationship between urinary and sexual health outcomes following prostate cancer treatment. Further investigation into these age-related patterns could help inform more personalized approaches to post-surgical care.

Although the model explained a modest proportion of the variability in urinary symptoms, the findings provide valuable insights into the importance of preoperative PSA levels and age. This preliminary analysis suggests that these factors could be considered in clinical decision-making. Future work with larger datasets and additional predictive features may improve the model’s performance and provide more accurate predictions.

### 4.1. Urinary Incontinence after Radical Prostatectomy

The surgical interventions employed for the management of prostate cancer encompass standard open retropubic radical prostatectomy, laparoscopic radical prostatectomy, and robot-assisted radical prostatectomy. During a radical prostatectomy, the patient’s pelvic floor muscles and the nerves regulating them may be damaged, leading to specific complications, with urinary incontinence (UI) being the most prevalent. UI in prostate cancer patients following surgical therapy is generally caused by anatomical or functional abnormalities in the urethral sphincter. This can involve harm to both the outer urethral sphincter and the related nerves, as well as an insufficient length of the functioning urethra. Despite ongoing improvements in surgical techniques, urinary function often changes following RP, with stress UI as a commonly reported adverse event in 20% to 87% of cases. The prevalence varies based on the definition utilized and the time point analyzed. The negative impact on the quality of life related to urinary function in men is well documented as a significant obstacle to engaging in social activities and physical exercise and can lead to mental health issues such as anxiety and depression [21,22,23].

### 4.2. Correlation between Sleep Quality and Urinary Incontinence

The findings indicate a moderate association between insomnia and urinary symptoms, which is notably significant at the starting point of the study and remains consistent throughout the several follow-up periods. The correlation matrix specifically demonstrates that there is a connection between poorer quality of sleep and more severe urinary symptoms at various stages following surgery. Baseline insomnia ratings exhibited a positive correlation with urinary symptoms, both at baseline and three months after the surgery. These findings indicate that patients who have more severe insomnia symptoms at the beginning of their recovery are more likely to experience more significant voiding difficulties as they continue to heal.

### 4.3. Correlation between Anxiety and Urinary Incontinence

Our analysis explored the potential relationship between anxiety and urinary incontinence (UI) across different time points. While the majority of the correlations were weak and not statistically significant, we found two noteworthy associations. The moderate positive correlation between baseline anxiety and urinary symptoms at 12 months (r = 0.46, *p* = 0.0037) suggests that higher anxiety levels at baseline may be predictive of worsening urinary symptoms in the long term. Additionally, the significant correlation between anxiety at 3 months and UI at 6 months (r = 0.34, *p* = 0.0373) highlights a potential short-term relationship between these variables. These findings suggest that anxiety may contribute to the progression of urinary symptoms in prostate cancer patients, and further research is needed to explore this connection more deeply.

### 4.4. Implications for Patient Care

The findings of this study are of major significance for the treatment of individuals with prostate cancer after undergoing surgery. Based on the discovered correlations, healthcare professionals should consider including sleep quality exams and interventions as part of the regular follow-up care for these patients. By taking proactive measures to manage insomnia, they may be able to reduce or alleviate UI, hence enhancing the overall QoL of patients. Furthermore, the incorporation of wearable technologies, such as Fitbit devices, presents a promising opportunity for ongoing surveillance of sleep patterns and other essential indicators. In addition to these measures, other established techniques for managing UI post radical prostatectomy can be implemented, thus contributing to improving patient outcomes. An RCT conducted by Azevedo et al. [24] showed the effectiveness of combining auricular acupuncture with pelvic floor muscle training in improving the QoL of patients with regard to urinary incontinence. Real-time data can offer healthcare providers significant insights into the patients’ healing process, allowing for more prompt and focused interventions. For instance, the early detection of sleep disruptions using wearable data could lead to therapies such as cognitive–behavioral therapy for insomnia (CBT-I), which has been proven to effectively enhance sleep quality [25,26,27].

What is more, the research methodology that we propose to enhance prostate cancer patient outcomes using AI techniques can be implemented in various medical fields. Islam et al. [28] presented a similar approach using data analytics in order to develop a precise stroke prediction model that can effectively assist in early intervention in this group of patients.

### 4.5. Need for Further Research

This study provides convincing evidence of the correlation between sleep quality and urine incontinence, but it also emphasizes the necessity for additional research. There are several areas that require further examination.
Longitudinal Studies: Future studies should focus on monitoring these patients for an extended duration to evaluate the durability of the identified correlations and to determine any long-term effects of sleep quality on urinary health.Interventional Studies: Randomized controlled trials are necessary to assess the effectiveness of sleep enhancement therapies specifically designed for men with prostate cancer. Conducting such studies could provide insights into whether enhancing the quality of sleep has a direct impact on reducing urinary symptoms.Mechanistic Studies: Gaining a comprehensive understanding of the fundamental mechanisms that link sleep issues with urinary symptoms could offer more profound insights. This involves investigating the impact of hormone fluctuations, stress, and other physiological factors that may influence this association [29].

### 4.6. Broader Context and Future Directions

The findings of this research are consistent with an increasing amount of research that highlights the interdependence of many facets of health. When it comes to treating prostate cancer, focusing on sleep quality is not just about achieving better sleep, but also about promoting overall recovery and quality of life. As the healthcare industry shifts towards a more holistic and patient-centered approach, incorporating research findings such as those from this study can result in more thorough and efficient treatment regimens [30,31].

Moreover, the utilization of artificial intelligence and sophisticated data analytics in the evaluation of health data, as demonstrated in the ASCAPE study, is a progressive strategy with significant potential. Artificial intelligence models trained on extensive datasets have the ability to forecast patient outcomes and suggest tailored interventions, therefore revolutionizing the provision of healthcare to patients. Subsequent investigations should persist in examining and enhancing these technologies, guaranteeing their ethical and efficient utilization for the betterment of patient well-being [32,33,34].

### 4.7. Limitations

The sample size of the current study might be an impediment for the generalization of the results. Therefore, including a wider range of patient populations can enhance the research by assessing the consistency of identified correlations across various demographic groups and therapeutic settings. This has the potential to improve the applicability of the results and contribute to the development of more tailored methods for patient treatment.

## 5. Conclusions

The ASCAPE project employs state-of-the-art machine learning techniques by combining powerful AI with thorough data collection methods, aiming to enhance the health-related quality of life of patients with prostate cancer after radical prostatectomy. By employing a rigorous study design and a strong methodology, the goal is to establish higher benchmarks in patient care and individualized treatment. This study employed sophisticated statistical techniques to investigate the intricate relationships between sleep quality and voiding issues in men with prostate cancer after surgical treatment. This study offered a comprehensive perspective on the factors that influence these crucial aspects of patient health by combining demographic and clinical data. The results indicate that strengthening the quality of sleep could potentially improve urinary disorders, therefore improving the quality of life of patients. In addition, by exploring the relationship between anxiety and UI, our results suggest that anxiety could worsen voiding issues in men with prostate cancer. Furthermore, our subgroup analysis showed that age-related variations could play a significant role in the relationship between urinary and sexual health outcomes following prostate cancer surgical treatment, as patients younger than 60 years of age presented with worse erectile function when associated with higher urinary symptoms. Nevertheless, further research is needed to thoroughly examine these associations and to formulate specific, efficient solutions. Through the ongoing integration of cutting-edge technologies and thorough data analysis, the healthcare sector can achieve substantial progress in enhancing the outcomes and quality of life for individuals diagnosed with prostate cancer.

## Figures and Tables

**Figure 1 healthcare-12-01817-f001:**
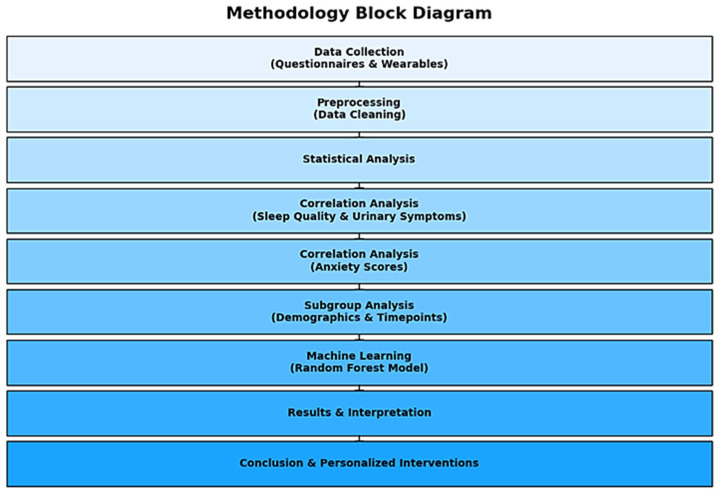
Methodology block diagram.

**Figure 2 healthcare-12-01817-f002:**
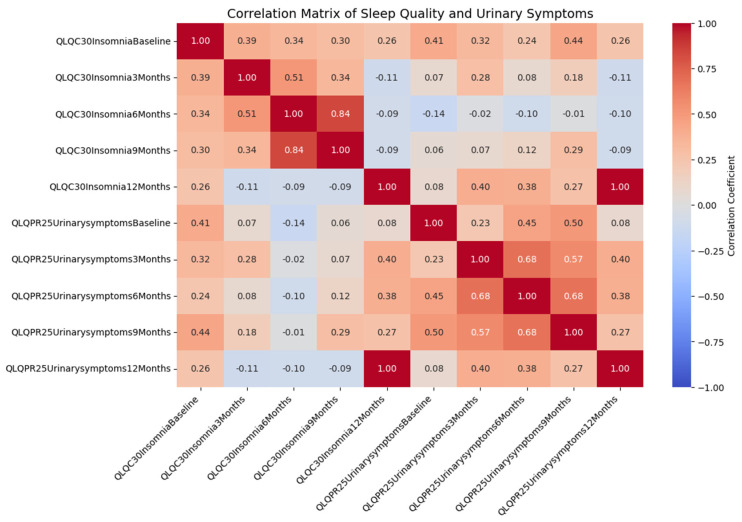
Correlation matrix of sleep quality and urinary symptoms, showing significant correlations at various time points.

**Figure 3 healthcare-12-01817-f003:**
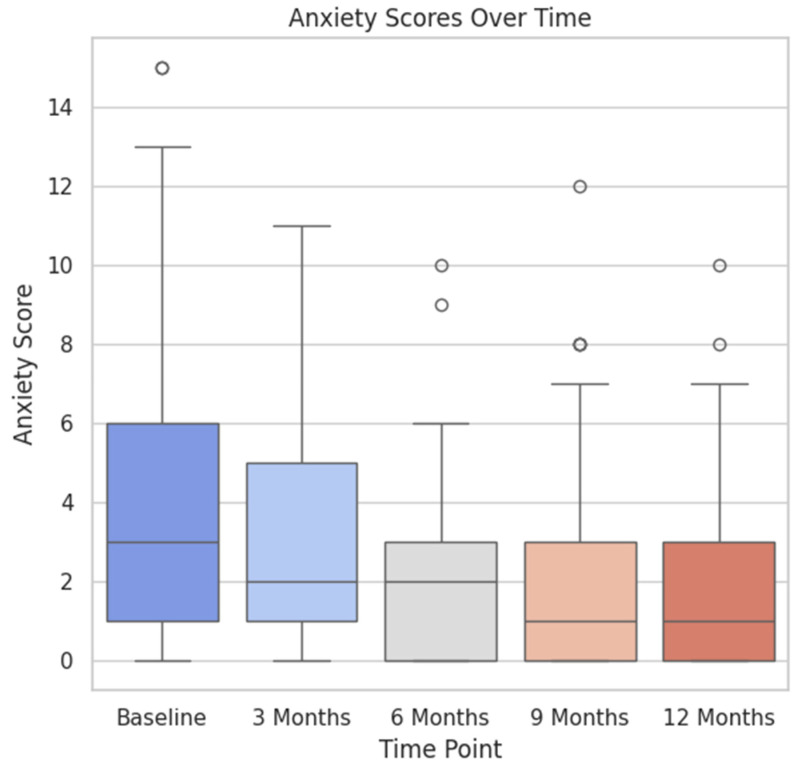
Anxiety scores over the specific time points.

**Figure 4 healthcare-12-01817-f004:**
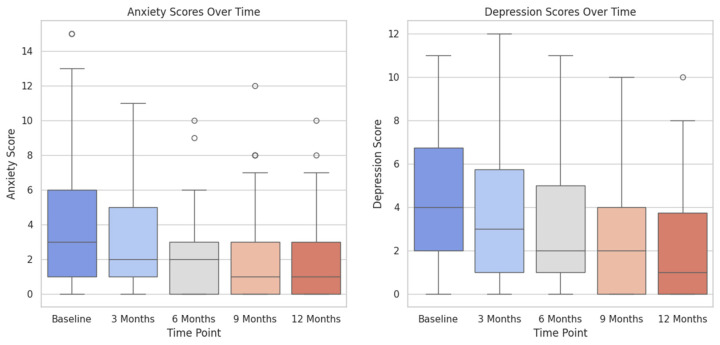
Box plot illustrating the depression scores over time, showing variability and occasional outliers across different time points.

**Figure 5 healthcare-12-01817-f005:**
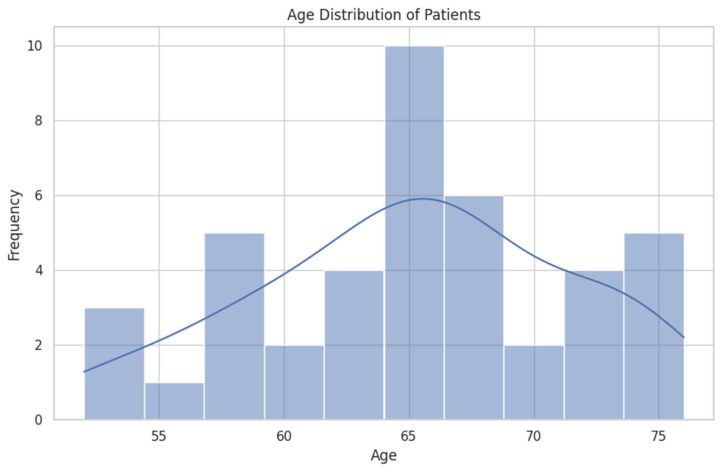
Histogram of the age distribution of patients.

**Table 1 healthcare-12-01817-t001:** Summary statistics of the demographic and clinical features of the study participants.

Feature	Mean (SD)	Median (IQR)
Age	65.1 (6.3)	66.0 (61.3–68.8)
Presenting Symptoms (LUTS)	0.3 (0.5)	0.0 (0.0–1.0)
Comorbidities	1.9 (1.3)	2.0 (1.0–3.0)
Drug Regimen	2.7 (2.3)	2.5 (1.0–4.0)
PSA Pre-op (ng/mL)	8.9 (7.2)	7.7 (5.7–9.0)
Gleason Score (Biopsy)	6.9 (0.8)	7.0 (6.0–7.0)
Blood Loss in OR (mL)	530.5 (149.5)	550.0 (450.0–650.0)
Length of Stay (days)	5.9 (1.9)	5.0 (5.0–6.0)
Duration of Surgery (min)	153.3 (32.4)	155.0 (130.0–170.0)
BCR	0.2 (0.4)	0.0 (0.0–0.0)

**Table 2 healthcare-12-01817-t002:** Key characteristics and functional results of the study participants across specified time points.

**Static Variables (Only at Baseline)**					
**Variable**			**Baseline (Mean ± SD)**		
Age (years)			61.54 ± 6.56		
Weight (Kg)			86.86 ± 11.41		
PSA (ng/mL)			9.11 ± 7.53		
Gleason Score (Biopsy)			6.92 ± 0.80		
**Time-Dependent Variables**					
**Variable**	**Baseline** **(Mean ± SD)**	**3 Months** **(Mean ± SD)**	**6 Months** **(Mean ± SD)**	**9 Months** **(Mean ± SD)**	**12 Months** **(Mean ± SD)**
Urinary Symptoms (QLQ-PR25)	21.4 ± 17.49	20.65 ± 18.61	15.31 ± 13.50	15.99 ± 16.51	11.40 ± 6.85
Erectile Function (IIEF)	39.11 ± 23.67	16.89 ± 10.91	18.41 ± 15.34	15.38 ± 13.27	15.59 ± 12.23

**Table 3 healthcare-12-01817-t003:** Statistically significant correlations between anxiety scores (measured by HADS) and urinary incontinence (UI) symptoms (measured by QLQ-PR25). This table shows moderate positive correlations at specific time points, indicating that higher anxiety levels are associated with worsening urinary symptoms.

Anxiety Time Point	UI Time Point	Correlation (r)	*p*-Value
HADSAnxietyBaseline	QLQPR25Urinarysymptoms12Months	0.46	0.0037
HADSAnxiety3Months	QLQPR25Urinarysymptoms6Months	0.34	0.0373

## Data Availability

The data presented in this study are available on request as they are not publicly available due to privacy and ethical restrictions.

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
