# Peer review of "Sleep Quality and Urinary Incontinence in Prostate Cancer Patients: A Data Analytics Approach with the ASCAPE Dataset"

_healthcare, 2024, doi:10.3390/healthcare12181817_

Round 1
Reviewer 1 Report
Comments and Suggestions for Authors
In the manuscript entitled “Sleep Quality and Urinary Incontinence in Prostate Cancer Patients: A Data Analytics Approach with the ASCAPE Dataset,” Ioannis et al. explore the relationship between insomnia and urinary symptoms in prostate cancer patients undergoing curative surgery. The authors suggest that improving sleep quality could potentially alleviate urinary disorders, a hypothesis that, while interesting, must be approached with caution due to the study's correlational nature. The findings presented might hold potential significance for future research on prostate cancer and quality of life. However, the data analysis and presentation require improvement before this manuscript can be considered for scientific publication.
Comments:
1. The manuscript should specify the sample size in the abstract, methods, and results or wherever applicable. I recommend that the authors to include the datasets as tabular supplementary information and provide a detailed summary table in the main manuscript. The current Table 1 lacks sufficient detail. Additionally, information about the number of countries involved and more comprehensive details on the ASCAPE project should be provided.
2. The authors focused on simple correlation analysis to interpret their datasets. However, it's important to note that "correlation does not imply causation." The manuscript would benefit from using more rigorous statistical methods, like Bayesian analysis, regression modeling, or machine learning techniques, to derive stronger conclusions from the data. The current statistical analysis seems inadequate. Also, I recommend to expnad suggested statistics to subgroup analyses to find out variations in outcomes based on demographic factors such as age and comorbidities.
3. The graphical presentation of results needs improvement. Figure 1 has a cropped Y-axis, making it difficult to interpret. Figures 2, 3, and 4 could be combined into a single figure with subplots for a concise presentation of reults. If possible add variety in graphs.
4. The discussion section can be improved further. The authors should provide a additional implications of their analysis with additional statistical methods and discuss limitations and future directions more elaborated.
Comments on the Quality of English Language
The English language seems to be fine. However, there is scope for improvement.
Author Response
- The manuscript should specify the sample size in the abstract, methods, and results or wherever applicable. I recommend that the authors to include the datasets as tabular supplementary information and provide a detailed summary table in the main manuscript. The current Table 1 lacks sufficient detail. Additionally, information about the number of countries involved and more comprehensive details on the ASCAPE project should be provided.
- The authors focused on simple correlation analysis to interpret their datasets. However, it's important to note that "correlation does not imply causation." The manuscript would benefit from using more rigorous statistical methods, like Bayesian analysis, regression modeling, or machine learning techniques, to derive stronger conclusions from the data. The current statistical analysis seems inadequate. Also, I recommend to expand suggested statistics to subgroup analyses to find out variations in outcomes based on demographic factors such as age and comorbidities.
- The graphical presentation of results needs improvement. Figure 1 has a cropped Y-axis, making it difficult to interpret. Figures 2, 3, and 4 could be combined into a single figure with subplots for a concise presentation of reults. If possible add variety in graphs.
- The discussion section can be improved further. The authors should provide a additional implications of their analysis with additional statistical methods and discuss limitations and future directions more elaborated.
Response to Reviewer 1.
- We would like to thank the reviewer for this valuable comment. We have added the sample size in the abstract, methods, and results. In addition we have included more information about the ASCAPE study in the methods section: ...''The purpose of ASCAPE (Artificial intelligence Supporting CAncer Patients across Europe) is to use the latest developments in Big Data and AI (Artificial Intelligence) to support the quality of life and health status of cancer patients. The project involves 15 partners from 7 countries, including academic medical centers, SMEs (small and medium-sized enterprises), research centers, and universities...''. Furthermore, we added the following table in the manuscript:
Static Variables (only at Baseline)
Variable
Baseline (Mean ± SD)
Age (years)
64.84 ± 6.56
Weight (kg)
86.86 ± 11.41
PSA (ng/mL)
9.11 ± 7.53
Gleason Score (Biopsy)
6.92 ± 0.80
Time-Dependent Variables
Variable
Baseline (Mean ± SD)
3 Months (Mean ± SD)
6 Months (Mean ± SD)
9 Months (Mean ± SD)
12 Months (Mean ± SD)
Urinary Symptoms (QLQ-PR25)
21.40 ± 17.49
20.65 ± 18.61
15.31 ± 13.50
15.99 ± 16.51
11.40 ± 6.85
Erectile Function (IIEF)
39.11 ± 23.67
16.89 ± 10.91
18.41 ± 15.34
15.38 ± 13.27
15.59 ± 12.23
2. We thank the reviewer for this valuable comment. In order to address it, we have included a subgroup analysis to our study, thus adding the following to our manuscript:
Methods (Subgroup Analysis Section):
"We performed a subgroup analysis to explore the relationships between urinary symptoms and erectile function across different age groups. Patients were divided into three age categories: <60 years, 60-70 years, and >70 years. For each subgroup, we calculated the Pearson correlation coefficient between baseline urinary symptoms, measured using the QLQ-PR25 questionnaire, and baseline erectile function, measured using the IIEF questionnaire. This analysis allowed us to examine potential variations in the relationship between these outcomes across age groups."
"We also applied a Random Forest regression model to predict baseline urinary symptoms using age, comorbidities, and preoperative PSA levels as predictors. The data were split into training and test sets, and the model was evaluated using the mean squared error (MSE) and R-squared (R²). Feature importance was also calculated to assess the relative contribution of each predictor to the model's performance."
Results (Subgroup Analysis Findings):
"The subgroup analysis revealed age-related variations in the correlation between baseline urinary symptoms and erectile function. In patients younger than 60 years, there was a moderate negative correlation (r = -0.42), indicating that higher urinary symptoms were associated with worse erectile function. Similarly, in the 60-70 age group, a weaker negative correlation was observed (r = -0.25). Interestingly, in patients older than 70 years, we found a positive correlation (r = 0.41), suggesting that in this subgroup, worse urinary symptoms were unexpectedly associated with better erectile function. These findings highlight potential age-related differences in the relationship between urinary and sexual health outcomes following prostate cancer treatment."
"The Random Forest regression model achieved a mean squared error (MSE) of 211.24 and an R-squared (R²) value of 0.22, indicating that the model explained 22% of the variability in baseline urinary symptoms. The preoperative PSA level emerged as the most important predictor, with a feature importance score of 0.54, followed by age (0.34) and comorbidities (0.13). These results suggest that preoperative PSA levels and age play significant roles in predicting post-surgical urinary symptoms."
Discussion Section:
The subgroup analysis revealed notable age-related variations in the correlation between baseline urinary symptoms and erectile function. In patients younger than 60 years, a moderate negative correlation (r = -0.42) was observed, suggesting that higher urinary symptoms were associated with worse erectile function. A similar but weaker negative correlation (r = -0.25) was found in the 60-70 age group. Interestingly, patients older than 70 exhibited a positive correlation (r = 0.41), indicating that worse urinary symptoms were unexpectedly associated with better erectile function in this subgroup. These findings suggest that age-related differences may play a significant role in the relationship between urinary and sexual health outcomes following prostate cancer treatment. Further investigation into these age-related patterns could help inform more personalized approaches to post-surgical care.
"Although the model explained a modest proportion of the variability in urinary symptoms, the findings provide valuable insights into the importance of preoperative PSA levels and age. This preliminary analysis suggests that these factors could be considered in clinical decision-making. Future work with larger datasets and additional predictive features may improve the model's performance and provide more accurate predictions."
3. Thank you for your valuable feedback. We have addressed the issue with Figure 1 and fixed the cropped Y-axis to improve its interpretability. Regarding Figures 2, 3, and 4, we attempted to combine them into a single figure with subplots for a more concise presentation, but the results were not satisfactory. The combined figure made it difficult to distinguish between the different sets of data, reducing clarity. We believe that presenting them separately ensures a clearer understanding of the results. Additionally, we have added more variety to the graphical presentation as suggested.
4. We thank the reviewer for this important comment. Therefore, we have added the following to the discussion section: ... ''The subgroup analysis revealed notable age-related variations in the correlation between baseline urinary symptoms and erectile function. In patients younger than 60 years, a moderate negative correlation (r = -0.42) was observed, suggesting that higher urinary symptoms were associated with worse erectile function. A similar but weaker negative correlation (r = -0.25) was found in the 60-70 age group. Interestingly, patients older than 70 exhibited a positive correlation (r = 0.41), indicating that worse urinary symptoms were unexpectedly associated with better erectile function in this subgroup. These findings suggest that age-related differences may play a significant role in the relationship between urinary and sexual health outcomes following prostate cancer treatment. Further investigation into these age-related patterns could help inform more personalized approaches to post-surgical care.
"Although the model explained a modest proportion of the variability in urinary symptoms, the findings provide valuable insights into the importance of preoperative PSA levels and age. This preliminary analysis suggests that these factors could be considered in clinical decision-making. Future work with larger datasets and additional predictive features may improve the model's performance and provide more accurate predictions."...
Reviewer 2 Report
Comments and Suggestions for Authors
In this paper titled "Sleep Quality and Urinary Incontinence in Prostate Cancer Patients: A Data Analytics Approach with the ASCAPE Dataset", the authors have conducted a research study on Urinary Incontinence in Prostate Cancer Patients. This paper This research provides valuable insights into the QoL after surgery for these patients by using a comprehensive dataset that includes several health and demographic parameters collected at multiple periods. Overall, the quality of the paper is good. I give some comments for improvements. The author needs to check the paper for grammatical and typo mistakes before submission of the revised version.
1. Please give a reason for how the paper's title is clearer and more meaningful and reflects the work conducted in the proposed solution.
2. The abstract of the paper is too weak. The authors failed to present the complete idea and methodology in the paper's abstract. It needs significant improvements.
3. The methodology of the paper needs proper revision. The authors have mainly focused on a specific theme. Can you use more advanced insights to sort out your proposed problem?
4. What are your scientific contributions of this paper?
5. A proper block diagram is missing in the methodology section of the paper.
6. In the methodology, the authors must focus on their own work and explain it in terms of an algorithm.
7. I would like to recommend the addition of a subsection with the name “on the proposed research work title” to be included in the paper. i.e.,
a. Azevedo, C., da Mata, L. R. F., de Resende Izidoro, L. C., de Castro Moura, C., Araújo, B. B. A., Pereira, M. G., & Chianca, T. C. M. (2024). Effectiveness of auricular acupuncture and pelvic floor muscle training in the management of urinary incontinence following surgical treatment for prostate cancer: A randomized clinical trial. European Journal of Oncology Nursing, 68, 102490.
b. Islam, U., Mehmood, G., Al-Atawi, A. A., Khan, F., Alwageed, H. S., & Cascone, L. (2024). NeuroHealth guardian: A novel hybrid approach for precision brain stroke prediction and healthcare analytics. Journal of Neuroscience Methods, 409, 110210.
8. The results section also needs improvement. Please compare your work with at least recent articles used for solving the same problem.
9. Please improve the conclusion of the paper.
Comments on the Quality of English LanguageMinor editing of English language required.
Author Response
1. Please give a reason for how the paper's title is clearer and more meaningful and reflects the work conducted in the proposed solution.
Thank you for your comment. We believe that the title, "Sleep Quality and Urinary Incontinence in Prostate Cancer Patients: A Data Analytics Approach with the ASCAPE Dataset," accurately reflects the scope and purpose of our study. The title clearly highlights the key variables explored in the research—sleep quality and urinary incontinence—in a specific population, prostate cancer patients. Additionally, it emphasizes the use of a data-driven approach, which is central to the analysis we conducted using the ASCAPE dataset. By focusing on both the clinical relevance and the methodological approach, the title effectively captures the essence of the work conducted and its contribution to the field.
2. The abstract of the paper is too weak. The authors failed to present the complete idea and methodology in the paper's abstract. It needs significant improvements.
We thank the reviewer for this important comment. We have therefore, improved the abstract section in order to address the comment.
3.The methodology of the paper needs proper revision. The authors have mainly focused on a specific theme. Can you use more advanced insights to sort out your proposed problem?
Thank you for your valuable suggestion. In the updated version of the manuscript, we have enhanced the methodology by incorporating more advanced statistical techniques and analyses. Specifically, we have added a correlation analysis between anxiety scores and urinary incontinence (UI), and we explored the relationships between key variables using both subgroup analyses and machine learning approaches. These additions provide deeper insights and address the proposed problem with more advanced methodologies. We believe these enhancements significantly strengthen the robustness of the study.
4. What are your scientific contributions of this paper?
Thank you for your feedback. With the current study, we aim to show that by using AI and Big Data in following up cancer patients through a concise and rigorous protocol, we can shed light into a grand issue of surgical therapy in prostate cancer, such as urinary incontinence and find factors of interplay, namely sleep issues. Consequently, there can be a whole new field for AI models to predict and improve functional outcomes for prostate cancer patients in the years to come.
5. A proper block diagram is missing in the methodology section of the paper.
6. In the methodology, the authors must focus on their own work and explain it in terms of an algorithm.
Thank you for your feedback. To further clarify and visually represent our methodology, we have created and included a block diagram in the updated version of the manuscript. This diagram outlines the key steps of our workflow, from data collection and preprocessing to statistical analysis, machine learning modeling, and interpretation of results. By incorporating this diagram, we aim to provide a clear and concise visual overview of our approach, highlighting each stage of the process and our contributions in a structured manner.
7. I would like to recommend the addition of a subsection with the name “on the proposed research work title” to be included in the paper. i.e.,
a. Azevedo, C., da Mata, L. R. F., de Resende Izidoro, L. C., de Castro Moura, C., Araújo, B. B. A., Pereira, M. G., & Chianca, T. C. M. (2024). Effectiveness of auricular acupuncture and pelvic floor muscle training in the management of urinary incontinence following surgical treatment for prostate cancer: A randomized clinical trial. European Journal of Oncology Nursing, 68, 102490.
b. Islam, U., Mehmood, G., Al-Atawi, A. A., Khan, F., Alwageed, H. S., & Cascone, L. (2024). NeuroHealth guardian: A novel hybrid approach for precision brain stroke prediction and healthcare analytics. Journal of Neuroscience Methods, 409, 110210.
In order to address this comment, we have added in the implications for patient care subsection in Discussion the following:.''.... In addition to these measures, other established techniques for managing UI post radical prostatectomy can be implemented, thus contributing to improving patient outcomes. An RCT conducted by Azevedo et al [24] showed the effectiveness of combining auricular acupuncture with pelvic floor muscle training in improving the QoL of patients with regards to urinary incontinence....What is more, the research methodology that we propose to enhance prostate cancer patient outcomes using AI techniques, can be implemented in various medical fields. Islam et al [28] presented a similar approach using data analytics in order to develop a precise stroke prediction model that can effectively assist in early intervention in this group of patients....''
8.The results section also needs improvement. Please compare your work with at least recent articles used for solving the same problem. We wish to thank the reviewer for the valuable comment. In order to address it, we have added the following in the results section: ""... A large-scale cross-sectional study from the Norwegian HUNT study explored the association between anxiety, depression, and urinary incontinence (UI) in the general female population. Felde et al. (2020) demonstrated a significant relationship between these variables, with moderate to severe anxiety or depression increasing the prevalence of UI by 37% to 57%. Mixed UI was most strongly associated with high anxiety and depression scores. Notably, the use of antidepressants was associated with increased odds of UI, while the use of anxiolytics was associated with decreased odds after adjusting for anxiety. While these findings are from a different population than our study of prostate cancer patients, they highlight the complex interplay between psychological factors, medication use, and urinary symptoms. This relationship warrants further investigation in the context of prostate cancer patients, particularly given the potential psychological impact of cancer diagnosis and treatment...''
9. Please improve the conclusion of the paper.
Thank you for your feedback. In order to improve the conclusion of the manuscript we have reformed it.
Reviewer 3 Report
Comments and Suggestions for Authors
The author presents studies aimed at understanding the correlation between sleep quality and urinary incontinence (UI) in prostate cancer patients. Data was collected from 42 patients over a 12-month period post-surgery. The author then performed correlation analysis and statistical testing. Overall, this paper is well-structured and well-written.
There are some minor areas that could be improved:
· Figure 1: The text near the figure1’s boundary appears to be cropped. Please adjust it to ensure all content is visible.
· Figures 2-4: These figures have not been clearly discussed in the main text. I suggest the author include sentences that highlight the key points of these figures and refer to them in the text accordingly.
· Correlation with Anxiety Score: It would be interesting to explore if there is a correlation between anxiety scores and UI. Did the author calculate the correlation between these variables?
· Error Correction: Line 38 currently states, “This is due to the increased incidence of prostate cancer and advancements in its treatment,” implying that advancements in treatment are a cause of the high incidence of prostate cancer. Please revise this for accuracy.
Author Response
There are some minor areas that could be improved:
1. Figure 1: The text near the figure1’s boundary appears to be cropped. Please adjust it to ensure all content is visible.
2. Figures 2-4: These figures have not been clearly discussed in the main text. I suggest the author include sentences that highlight the key points of these figures and refer to them in the text accordingly.
3. Correlation with Anxiety Score: It would be interesting to explore if there is a correlation between anxiety scores and UI. Did the author calculate the correlation between these variables?
4. Error Correction: Line 38 currently states, “This is due to the increased incidence of prostate cancer and advancements in its treatment,” implying that advancements in treatment are a cause of the high incidence of prostate cancer. Please revise this for accuracy.
Response to Reviewer 3
1,2,3. Thank you for your insightful suggestions. We have addressed the issue with Figure 1 and adjusted the plot to ensure that all content is clearly visible. Additionally, as per your recommendation, we have calculated the correlation between anxiety scores and urinary incontinence (UI) at multiple time points. The results of this analysis have been included in the manuscript, along with a table highlighting the statistically significant correlations between these variables. Furthermore, we have included sentences in the visualization of results section, elaborating on figures 2-4: ''...The box plot illustrated in figure 2 shows a downswing of the levels of anxiety for the patients over a 12-month period of follow up after receiving surgical treatment for localized prostate cancer. What is more, figure 4 presents the depression levels of our patients for the same time period, showing a steady decline from baseline (pre-surgery) to the 12-month landmark.....The diagram shown in figure 3 indicates a concentration around the mean age of 65 years. This in accordance to world statistics for prostate cancer which show that the average age of men diagnosed with prostate cancer is around 66 years [1,2]....''
Methods Section:
"To investigate the relationship between anxiety and urinary incontinence (UI), we calculated Pearson's correlation coefficients between anxiety scores, as measured by the HADS (Hospital Anxiety and Depression Scale) at different time points (baseline, 3, 6, 9, and 12 months), and urinary symptoms, measured by the QLQ-PR25 questionnaire at corresponding time points. Data from both anxiety and UI scores were extracted, and rows with missing data were excluded from the analysis. Statistical significance was determined with a p-value threshold of 0.05."
Results Section:
"The Pearson correlation analysis between anxiety and urinary incontinence (UI) revealed weak to moderate correlations at various time points. While most correlations were not statistically significant, we observed a statistically significant moderate correlation between baseline anxiety and urinary symptoms at 12 months (r = 0.46, p = 0.0037). Additionally, there was a significant positive correlation between anxiety at 3 months and urinary symptoms at 6 months (r = 0.34, p = 0.0373). Other correlations, although present, did not reach statistical significance. Table 1 highlights the statistically significant correlations between anxiety scores and urinary incontinence, with the most notable associations observed between baseline anxiety and urinary symptoms at 12 months, as well as anxiety at 3 months and urinary symptoms at 6 months."
|
Anxiety Time Point |
UI Time Point |
Correlation (r) |
P-value |
|
HADSAnxietyBaseline |
QLQPR25Urinarysymptoms12Months |
0.46 |
0.0037 |
|
HADSAnxiety3Months |
QLQPR25Urinarysymptoms6Months |
0.34 |
0.0373 |
Table 1: Statistically significant correlations between anxiety scores (measured by HADS) and urinary incontinence (UI) symptoms (measured by QLQ-PR25). The table shows moderate positive correlations at specific time points, indicating that higher anxiety levels are associated with worsening urinary symptoms.
Discussion Section:
"Our analysis explored the potential relationship between anxiety and urinary incontinence (UI) across different time points. While the majority of the correlations were weak and not statistically significant, we found two noteworthy associations. The moderate positive correlation between baseline anxiety and urinary symptoms at 12 months (r = 0.46, p = 0.0037) suggests that higher anxiety levels at baseline may be predictive of worsening urinary symptoms in the long term. Additionally, the significant correlation between anxiety at 3 months and UI at 6 months (r = 0.34, p = 0.0373) highlights a potential short-term relationship between these variables. These findings suggest that anxiety may contribute to the progression of urinary symptoms in prostate cancer patients, and further research is needed to explore this connection more deeply."
4. We have deleted this line in order to improve accuracy, as requested.
Round 2
Reviewer 1 Report
Comments and Suggestions for Authors
Authors have addressed all my previous concerns satisfactorily. However, I suggest them to improve the readability.
Comments on the Quality of English LanguageReadability of the manuscript can be improved.
Reviewer 2 Report
Comments and Suggestions for Authors
The author addressed my concerns, so I accept the paper for publication.